# Evaluation of the Effectiveness of a Nordic Walking and a Resistance Indoor Training Program: Anthropometric, Body Composition, and Functional Parameters in the Middle-Aged Population

**DOI:** 10.3390/jfmk8020079

**Published:** 2023-06-15

**Authors:** Alessia Grigoletto, Mario Mauro, Stefania Toselli

**Affiliations:** 1Department of Biomedical and Neuromotor Sciences, University of Bologna, Via Selmi 3, 40126 Bologna, Italy; alessia.grigoletto2@unibo.it; 2Department for Life Quality Studies, University of Bologna, 47921 Rimini, Italy

**Keywords:** anthropometric characteristics, body composition, indoor training, Nordic walking, outdoor training, physical activity, physical test

## Abstract

Sedentary behaviors are increasing in the population, so strategies for the increment of physical activity levels are needed. The use of green space seems to be a valid support to be more active. The present study aimed to compare the effectiveness of a period of outdoor training (Nordic walking (NW)) with indoor resistance training (GYM) in a nonclinical population based on anthropometric characteristics, body composition, and functional parameters. This study was conducted on 102 participants (77 middle-aged people performed NW and 25 performed indoor training). Participants were measured twice: at baseline and after three months. Anthropometric measurements (weight, BMI, skinfolds, perimeters), body composition, bioelectrical impedance, vectorial analysis (BIA and BIVA), and physical tests were carried out. A two-way repeated measures analysis of variance (ANOVA) was performed to evaluate the effect of the treatments, groups, and sexes. There were several intervention effects linked to a decrease in fat parameters (such as skinfolds, fat mass, and percentage of fat mass). Considering the type of intervention, NW showed a higher increase in muscle mass and a higher decrease in fat parameters than the GYM group. In conclusion, the two types of training could represent a good way to remain active and prevent sedentary behaviors.

## 1. Introduction

Physical inactivity is one of the most relevant sedentary behaviors, which causes massive effects on the public health global economy [1,2,3]. Participation in regular physical activity (PA) can significantly reduce the risk of developing cardiovascular disease, stroke, sarcopenic obesity, cancers, and diabetes, and improve mental health outcomes, such as depression and anxiety [4,5]. In addition, participation in PA is useful for maintaining and slowing physiological age predicated on the decline of the musculoskeletal system [6,7]. While the importance of PA is well-established, a significant proportion of the adult population remains inactive [8]. In Italy, despite the evidence, only 31% of adults declared to have a physically active job [9]. Di Bonaventura et al. (2018) reported that 12.89% of Italian people were obese (9.49% were obese class I, 2.28% were obese class II, and 1.12% were obese class III) [10]. Several factors influence participation in PA, and greater attention has recently focused on the role of the environment in promoting PA [11,12]. Green space seems to be an optimal environment for exercise, due to the safety, accessibility, and attractiveness of these places [13]. Several observational studies have searched to establish whether a relationship between green space and PA exists [11]. However, studies in this area are still lacking and are far from conclusive [14]. A systematic review demonstrated the paucity of high-quality evidence in the studies carried out so far and the necessity for further research in this area [12]. Thus, presently, it is not possible to establish if PA carried out in green spaces is more effective than indoor PA in producing physical, physiological, and motor changes in participants.

To achieve this purpose, the analysis of body composition can provide useful information since it is an indicator of health, nutritional status, and functional capacity [15]. In the last years, the analysis of body composition by bioelectrical impedance analysis (BIA) has become one of the most used methods, due to its easy use, precision, and accuracy [16,17,18]. By employing the bioimpedance-based predictive equation, it is possible to estimate body composition parameters, such as fat mass, muscle mass, and body water, and monitor their change [19]. By the qualitative approach, it is also possible to estimate body composition through the raw bioimpedance parameters (resistance (R) and reactance (Xc)) as a point on the R–Xc graph, in which length and slope are considered. The vector slope indicates the extracellular/intracellular (ECW/ICW) ratio and the integrity of the cell membrane [19,20]. In addition, some studies have shown how the phase angle (PhA), which is an indicator of health status, can be influenced and modulated by exercise [21,22,23]. An inverse relationship between PhA and inflammatory biomarkers [21] and a positive association with cellular health has been reported [22,23].

Body composition improvements are important aspects to take into account when we try to identify what could be the best type of activity for the middle-aged population. Performing PA in green urban spaces has several important benefits, but usually, it is performed with lower intensity than indoor PA. For this reason, we decided to compare indoor and outdoor PA performed with the same intensity to better understand if one could have a greater impact on anthropometric and body composition characteristics.

To our knowledge, no studies compare the effectiveness of indoor and outdoor PA on body composition and physical parameters in Italy. Nordic walking (NW) is an easy type of PA that is usually proposed for the clinical population and the elderly, but that also shows several potential benefits for nonclinical populations [24]. Resistance training is a kind of PA that can maintain good health conditions and reverse the adverse effects of ageing on cellular integrity and function. Both these kinds of PA are usually suggested for the middle-aged population to remain active. For this reason, the present study aimed to evaluate the efficacy of a period of three months of training outdoors (in particular, NW) and in GYM (resistance training) environments on a healthy middle-aged population.

## 2. Materials and Methods

### 2.1. Study Design and Participants

This is an intervention study design that comprehended 3 months of PA (NW as outdoor PA and resistance training as indoor PA) and two measurements (at baseline and after the training program). Recruitment occurred thanks to two sports society: “Nordic Walking Italy”, specifically their headquarters in Venice, and “Arca”, based in Mirano (VE). The two sports societies made announcements to recruit people to participate in their normal activities, and they also explained the possibility of participating in the present study. To increase participation and adherence in the study, each adult could select the NW or GYM activity. Nordic Walking Italy conducts activities throughout the province of Venice, in city parks, along the banks, and always in the open air. They manage different walking groups in Mestre, Marghera, Spinea, and Martellago. Conversely, Arca is based in Mirano (VE) and conducts activities in the school gym of the municipality. The exclusion criteria were (a) have a chronic disabling disease, being bedridden, institutionalized, or hospitalized, (b) not being independently mobile, i.e., not requiring human assistance or the aid of devices such as crutches, walkers, etc., and (c) having amputations, pacemakers, or the presence of a chronic metabolic disease.

Before starting the participants’ enrollment, the sample size estimation for the repeated measures ANOVA F test for between–within groups with a Greenhouse–Geisser correction was assessed. The study parameters included were as follows: α = 0.05, 1 − β = 0.80, number of groups = 2, number of repeated measures = 2, between–within variance explained = 0.05, correlation = 0.25, and error variance = 2; the estimated sample size was 120. After a preliminary explanation of the study protocol, a total of 135 participants were enrolled in the study. Fifteen participants did not meet the inclusion criteria and were excluded before the protocol began (Figure 1). Three adults who chose NW and fifteen people who preferred GYM activities did not complete the period of training, so they were excluded from the study. Therefore, the sample was finally composed of 102 participants who performed both the measurements (before and after the training period). NW participants numbered 77, and the GYM group was composed of 25 people. All participants signed an informed consent to participate in the study. The study was approved by the Bioethics Committee of the University of Bologna (prot. N. 022254).

### 2.2. Intervention Training Programs

Participants were engaged in two training sessions of about 60 min each, two times a week. Every training session was composed of 10 min of warm-up, 45 min of the main part, and 5 min of stretching exercise. For the NW training, three instructors followed the groups in different parks, proposing the same kind of training to the different groups, with the same kind of intensity. For the GYM training, the same instructors followed the group and proposed several resistance exercises. For both the NW and GYM groups, instructors proposed 15 min of upper limb exercises and 10 min of core and stability exercises. In the NW training, both upper limb and core exercises were performed with body weight resistance, during walking (upper limb) or in a lying position (core). The upper limb exercises were side and front risers, rows, intra and extra rotations for the shoulders, and military presses. The core exercises included different versions of planks, crunches, and sit ups. Each exercise was repeated for 45 s, with a 15-s rest, two times. For the remaining 20 min of the main part, the NW group performed walking exercises at various speeds and inclinations, while the GYM group performed lower-limb resistance exercises, such as squats, glute bridges, rear and side lunges, and deadlifts with one or two legs. Each exercise was repeated for 45 s, with a 15-s rest, two times. All the resistance exercises were executed with elastic bands of different resistances represented by colors (red, yellow, green, blue, and black). 

To control the training intensity, instructors taught the participants to use the rating of perceived exertion scale (RPE, CR-10). Instructors collected participants’ RPE before, during, and at the end of the training. The intensity level was set from 5 to up to 8 on the RPE scale. RPE before exercises was detected to ensure that each participant was able to perform the training with no risk. During exercise, RPE was used to induce desired stimuli; when a participant perceived an exertion lower than 5, instructors encouraged them to increase the exercise intensity (speed, inclination, or band), whereas if it was higher than 8, the participant was directed to slow down. Thirty minutes after the end of a training session, RPE was recorded to calculate the training load (TL = RPE × minutes of training) [25].

### 2.3. Anthropometric Characteristics

Participants’ anthropometric measures were recorded twice: at the baseline and after the three months of training. Height was recorded with a standing stadiometer (GPM, Steckborn, Switzerland), while a calibrated electronic scale (Seca, Basel, Switzerland) was used to measure the body mass. Body mass index (BMI) was calculated as the ratio of body weight to height squared (kg/m^2^), and the WHO cutoff was used to estimate the weight status of the subjects: a BMI value less than 18.5 was classified as underweight, from 18.5 to 24.9 was considered normal weight, from 25 to 29.9 was overweight, and more than 30 was classified as obese [26]. In addition, circumference (relaxed and contracted arm, waist, hip, and calf) and skinfold (biceps, triceps, subscapular, suprailiac, supraspinale, lateral, and medial calf) measurements were carried out. Circumferences were taken using a nonstretchable tape measure (GPM, Steckborn Switzerland), and skinfolds were measured with a skinfold caliper (Lange, Beta Technology, Cambridge, MD, USA). The total upper arm and calf area, upper arm and calf muscle area, and upper arm and calf area were calculated [27]. Body density was calculated using the Durnin et al. equations, and then body composition parameters were estimated using Siri’s converting equation [24,28]. All the anthropometric measurements were carried out by the same operator, specifically trained according to a standardized protocol [29,30,31].

### 2.4. Body Composition

The impedance measurements were performed with a bioimpedance analyzer (BIA 101 Anniversary, Akern, Florence, Italy) at a frequency of 50 kHz. The accuracy of the BIA instrument was validated before each test session, following the manufacturer’s instructions. The participants were assessed in the supine position with legs abducted 45° compared to the median line of the body and arms abducted 30° from the trunk. After cleansing the skin with alcohol, two electrodes were placed five centimeters apart from each other on the right hand and two on the right foot. Participants were instructed to abstain from food and liquids for at least four hours before the test, to urinate about 30 min before the measures, to not consume alcoholic beverages for at least 48 h, and to avoid the use of diuretics at least seven days before each assessment. Vector length was calculated as (adjusted R2 + adjusted Xc2) 0.5 and PhA as the arctangent of (Xc/R) × (180°/π). Bioimpedance vector analysis was carried out using the BIVA method, normalizing the R and Xc parameters for height (h) in meters. Bioelectrical-specific values for women were used as a reference to build the tolerance ellipses on the R–Xc graph [32]. R, Xc, and PhA were plotted in the BIVA as a vector within a specific tolerance ellipse (specific profile for sex and age); this allowed the evaluation of soft tissue through patterns based on percentiles of their electrical characteristics [33]. A BIVA vector that falls in the 50% tolerance ellipse represents a normal tissue impedance, while a vector in the 75% tolerance ellipse displays an abnormal tissue impedance. BIVA outcomes could be analyzed through the vector direction to the y and *x*-axis. Horizontal displacement means changes in soft tissue mass (less soft tissue to the right pole and more soft tissue to the left pole); vertical displacements show changes in the hydration of the tissue (hyperhydration in the short vectors and dehydration in the long vectors) [33].

### 2.5. Physical Test

Three physical tests were performed by all participants. To avoid any confounding effect of time of day [34], all test sessions were performed in the morning, both at the baseline and after three months.

To evaluate the strength of the hands, a dynamometer (Takei Scientific Instrument Co., Niigata City, Japan) was used both for the right and left handgrip. Participants were in a sitting position at a 90-degree flexion of their elbow and performed three trials with a 1 min rest period between each test. The highest value of all three measurements was used for analysis. 

A chair-stand test (squat test) was executed to assess the strength and endurance of the lower limbs. Every test was preceded by an explanation and demonstration of the test. People were allowed one practice trial before the actual measurements. The same standard chair without armrests was used for all the participants. Participants were instructed to sit in the middle of the chair, back straight, feet approximately shoulder-width apart, and placed on the floor at an angle slightly back from the knees, with one foot slightly in front of the other to help maintain balance when standing. The instructions to participants were to stand up and sit down again as many times as possible for 30 s. Participants were encouraged to continue to sit and stand throughout the test. The number of repetitions was recorded, and it represented the unit for this measure [35].

The 6-Minute Walking Test is a simple test to measure exercise capacity [36]. It was explained that the participants should have to walk for six minutes, and they were instructed to follow their gait and to slow down or stop if they became fatigued, but to resume once able. Each time the subject passed the starting position a lap was recorded. Using an even-toned encouraging phrase, the time remaining in the test was reported to the participants at one-minute intervals. The timer was not stopped if the participants needed to rest. Once the six minutes concluded, the participants were instructed to stop and remain stationary while the endpoint was marked. Once marked, the total distance walked was calculated in meters [37].

### 2.6. Statistical Analysis

The analysis was performed with Stata software for Windows 10, version 17 (publisher: StataCorp, 2021. Stata Statistical Software: Release 17. College Station, TX, USA, StataCorp LP). Descriptive analyses were performed, and each result was reported as the variable mean ± standard deviation (SD) at two different times (baseline and after three months of PA). To check the normal distribution of the variables, a Shapiro–Wilk test was carried out. A transformation function (natural logarithm) was applied to reduce curve skewness if variable data did not distribute as a Gaussian curve. A two-way repeated measures analysis of variance (ANOVA) was computed to compare groups, considering treatment, time, and sex effects. To meet the GLM sphericity assumption, a Greenhouse–Geisser correction was then applied. A Hotelling’s test was performed to observe eventual differences in the BIVA representation.

Statistical significance was set at *p* < 0.05.

Finally, the statistical power achieved was calculated with the sample size = 102, number of groups = 2, number of repeated measures = 2, between–within variance explained = 0.05, error variance = 1.5, and correlation = 0.2: 1 − β = 0.823.

## 3. Results

### 3.1. Training Load of the Participants 

Table 1 reports the TL of the participants in the two different training programs.

There were no significant differences between the TL of the two training programs. 

### 3.2. Baseline Characteristics of the Participants

The largest sample was comprised of females (70, 69.3%) with a mean age of 56.96 ± 6.64 years. Men who participated in the study (31, 36.7%) were older than women (61.30 ± 8.37 years). 

Table 2 reports the characteristics of the sample divided by the kind of training performed (NW or GYM activity). 

### 3.3. Effects of Three Months of Training

Table 2 shows several significant differences induced by the intervention, by the two types of training, and by sex.

Regarding the effects of the interventions, the biceps, subscapular, and suprailiac skinfolds showed a significant decrease. At the same time, waist circumference, %F, FM, and CFA presented a significant decrease. Conversely, arm-relaxed circumference, phase angle, number of squats, TUA, and UMA showed a significant increase after the period of training.

Considering the comparison of the two intervention groups, people who practiced NW had a greater decrease in weight, triceps and lateral calf skinfolds, hip circumference, UFA, and CFA. On the contrary, the value of the arm-relaxed circumference showed a significant increase in the NW group in comparison with the GYM group.

The calf circumference, number of squats, and TCA presented a reverse trend. People in the GYM group showed a significant increase in these variables.

Considering the differences linked to sex, men had significantly higher value for the phase angles and a significantly lower value for reactance than women.

### 3.4. Effects of NW and GYM Training on BIVA

Figure 2 shows the BIVA ellipses of the NW and GYM groups at baseline and follow-up: no significant changes appeared between the two groups (*p* = 1.1 in the pre-analysis and *p* = 0.6 in the post-analysis).

Figure 3 shows a significant difference in the impedance value in the GYM group. In particular, the reactance had a significant decrease. The reactance is a measure of cellularity integrity, and the decrease could be a symptom of inflammation. The NW group shows a reduction of resistance and an increase in reactance, but these results are not significant.

## 4. Discussion

The present study aimed to evaluate the effectiveness of a period of three months of outdoor (NW) and GYM (resistance) training in healthy middle-aged people. To our knowledge, there are no studies that consider the comparison between the two types of training (NW and GYM) on body composition and physical motor characteristics. NW has gained popularity worldwide as a health-promoting activity [38]. NW could be considered a total-body version of walking, with greater body muscle activity due to the use of poles and potentially enhanced physical fitness benefits [39,40]. NW has several important benefits for the population, such as resting blood pressure and heart rate, increasing exercise capacity, quality of life, and maximal oxygen consumption [41]. Resistance training is one of the most used types of PA, and in addition, it has the potential to help people maintain a good health condition and reverse the adverse effects of ageing on cellular integrity and function [42]. In fact, authors reported improvements in physical performance, movement control, walking speed, and functional independence, which may assist the prevention and management of several chronic diseases [43]. In addition, resistance training improved cardiovascular health and increases bone mineral density [44]. Although the two types of PA could potentially have positive effects after a period of training, further research is needed to reveal the optimal dose–response relationship [45]. 

In the present study, considering the whole sample, the three months of NW and GYM training showed positive effects: in both activities, a significant decrease in FM and %F and a significant increase in the squat test were observed. This represents a positive aspect, which could be linked to an increase in the strength of the lower body and to an increase in the capacity of resistance. This is partially in contrast with a previous study about NW which assumed that the use of poles, used as a support, reduced the training effect on lower extremities [46]. Having more strength and resistance could drive people to continue PA for a longer period and thus be more active [31]. The previous study also showed a significant increase in handgrip strength, in contrast with the results of the present study [46,47]. Maybe the period of three months was too short to observe a significant change in upper-body muscle strength. Both the intervention groups showed an increase in PhA. The increase in this parameter is generally associated with an increase in strength (in this case, of the lower body) and an alteration in cellular membrane integrity or body fluid, or a combination of both [48]. This aspect is important because muscle strength is recognized as a good predictor of adverse health outcomes [49]. This result indicates the promotion of additional positive effects on cellularity cell size and integrity in the cell membrane. The PhA changes highlighted are in line with the results of previous studies [50]. Several studies have considered PhA and its relationship with health status, and it is an important factor to prevent the ageing process [48]. The BIVA changes may be due to changes in nutritional habits during the intervention period, an aspect that has not been monitored and could have caused disparity changes in R and Xc, as reported in previous studies [49]. The present results partially contrast with a previous study, which found significant improvement in BIVA and fat parameters after a resistance training period [50]. This could also be linked to the period of training; Fukuda et al. (2016) found a significant increase in BIVA parameters after a period of six months of resistance training [50]. So, probably, three months of training is too short to observe significant changes in BIVA parameters.

In addition, the whole sample of NW showed a significant decrease in other parameters, such as hip circumference. Regarding the comparison between the two groups, the NW group seemed to have more positive effects. NW had a greater decrease in several parameters connected to body fat (triceps skinfold, UFA, and CFA).

A previous study compared the effects of conventional walking, NW, and resistance training in older adults [51]. The results showed that walking improves cardiorespiratory fitness, resistance exercise improves muscle strength, and NW is a combination of the two types of activities and provides improvement to both components [51]. Performing NW takes less time than performing the same amount of walking with additional resistance training sessions. Experimental research suggested that performing the activity in nature has additional benefits in comparison to performing it in an indoor environment [51], and in addition, exposure to nature could provide restoration from stress and mental fatigue [52]. A greater long-term adherence to exercise participation was shown in outdoor PA by some studies than in GYM exercise interventions [49,50,51,52]. The present study has some limitations that should be addressed. The period of training of three months could be considered relatively short. It could be interesting to extend the period of training to six months, or a year, to determine if the results could be different. In addition, the sample size was limited; it could be interesting for future research to include more participants. Furthermore, participants were not asked for information about diet and alimentary behavior. So, for future research, it will be an important aspect to consider.

## 5. Conclusions

The present study aimed to evaluate the benefits of a period of NW and indoor resistance training in the general middle-aged population to understand the effectiveness of the two types of PA. Both NW and indoor training have positive effects on the participants. NW is usually proposed for specific kinds of populations or the rehabilitation of chronic disease, although it seems to have several potential benefits also for the general population. However, the popularity of this sport is increasing, and it would be important to encourage its practice in the population to promote a decrease in sedentary behavior.

## Figures and Tables

**Figure 1 jfmk-08-00079-f001:**
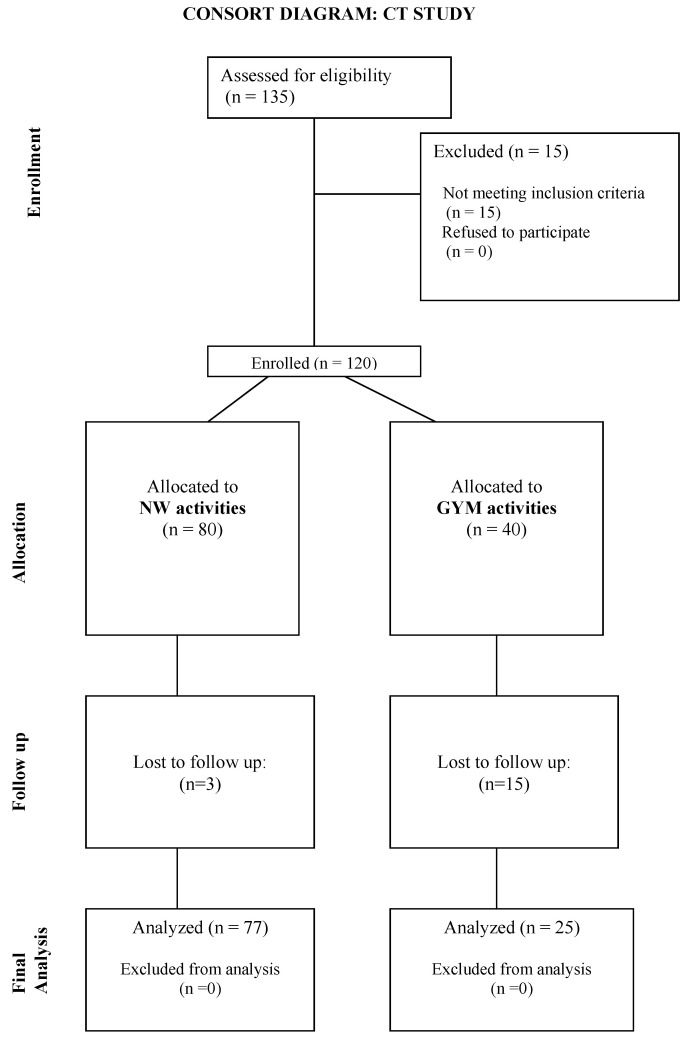
Participant flowchart.

**Figure 2 jfmk-08-00079-f002:**
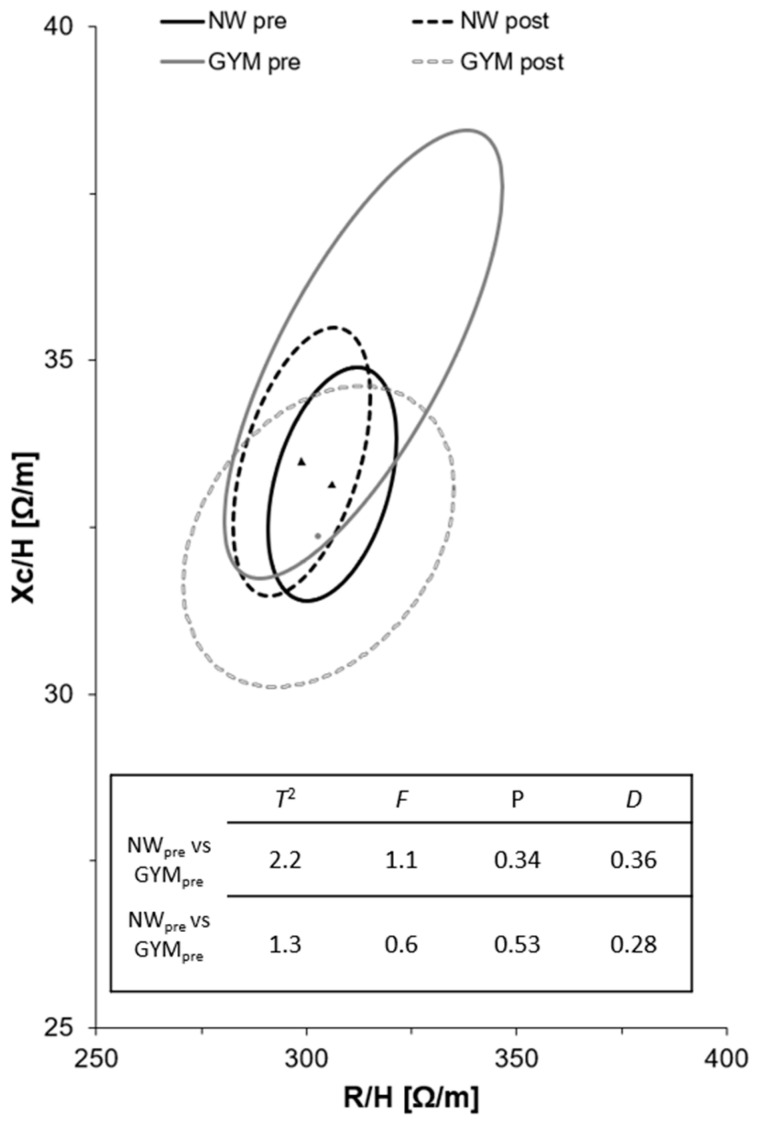
R/H-Xc/H and paired graphs for the multivariate changes in classic resistance and reactance in the GYM and NW groups. Mean vector displacements with 95%, confidence ellipses, and results of the Hotelling’s T^2^ test are shown. Note: triangles represent NW mean points and dots represent GYM mean points.

**Figure 3 jfmk-08-00079-f003:**
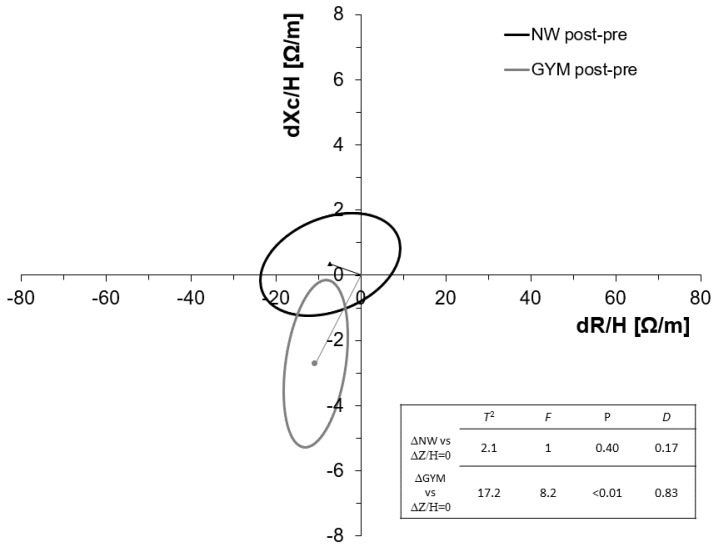
Paired BIVA graph on impedance delta among the NW and GYM groups. Note: triangles represent NW mean points and dots represent GYM mean points.

**Table 1 jfmk-08-00079-t001:** TL of the participants in the two different training programs.

Week	Day	NW (Mean ± SD)	GYM (Mean ± SD)	F	*p*
Week 1	Day 1	395.36 ± 58.09	402.50 ± 57.28	0.02	0.052
	Day 2	387.86 ± 51.12	397.50 ± 55.42	0.25	0.085
Week 2	Day 1	390.00 ± 67.69	382.50 ± 55.42	0.56	0.753
	Day 2	371.79 ± 56.54	397.50 ± 58.18	1.46	0.652
Week 3	Day 1	381.43 ± 50.43	367.40 ± 36.74	4.46	0.123
	Day 2	397.50 ± 63.25	405.00 ± 56.65	0.05	0.257
Week 4	Day 1	395.36 ± 63.47	395.00 ± 63.59	1.26	0.951
	Day 2	394.29 ± 64.58	397.50 ± 60.81	1.36	0.746
Week 5	Day 1	401.79 ± 57.12	382.30 ± 55.42	2.65	0.563
	Day 2	397.50 ± 56.70	397.20 ± 57.36	0.08	0.452
Week 6	Day 1	380.36 ± 51.52	397.80 ± 60.81	3.20	0.874
	Day 2	401.79 ± 57.12	405.00 ± 55.66	0.07	0.658
Week 7	Day 1	394.29 ± 64.58	395.00 ± 65.39	5.20	0.887
	Day 2	397.50 ± 65.70	395.30 ± 68.10	3.21	0.632
Week 8	Day 1	380.86 ± 52.51	372.50 ± 43.26	2.03	0.358
	Day 2	409.71 ± 57.12	406.00 ± 56.65	1.02	0.742
Week 9	Day 1	380.36 ± 25.16	382.50 ± 55.42	4.25	0.896
	Day 2	407.91 ± 67.50	390.00 ± 30.65	0.02	0.554
Week 10	Day 1	397.50 ± 66.50	405.50 ± 56.58	0.04	0.665
	Day 2	394.27 ± 64.58	397.56 ± 55.42	0.10	0.578
Week 11	Day 1	380.36 ± 51.52	382.50 ± 55.58	3.89	0.832
	Day 2	397.00 ± 57.60	395.63 ± 63.59	4.02	0.752
Week 12	Day 1	401.79 ± 57.12	395.00 ± 59.63	0.66	0.348
	Day 2	394.29 ± 64.58	397.50 ± 60.81	0.59	0.658

**Table 2 jfmk-08-00079-t002:** Anthropometric characteristics, body composition, and physical test values for groups.

	♀ (n = 70)				♂ (n = 31)				ANOVA Comparisons					
	GYM (n = 14)		NW (n = 56)		GYM (n = 10)		NW (n = 21)		Intervention Effect		Type Effect		Sex Effect	
Var	Pre	Post	Pre	Post	Pre	Post	Pre	Post						
	Mean (SD)	Mean (SD)	Mean (SD)	Mean (SD)	Mean (SD)	Mean (SD)	Mean (SD)	Mean (SD)	F	*p*	F	*p*	F	*p*
Weight, kg	65.65 (11.69)	65.34 (12.05)	59.88 (10.69)	63.41 (14.00)	84.58 (8.99)	83.86 (7.53)	78.40 (7.06)	76.20 (7.80)	0.01	0.937	4.14	0.045 *	0.06	0.799
BMI, kg/m^2^	24.01 (3.81)	23.80 (3.89)	22.54 (3.57)	23.96 (5.63)	27.19 (3.02)	26.97 (2.81)	24.99 (1.66)	24.29 (1.98)	0.02	0.881	2.61	0.109	1.42	0.236
Triceps SK, mm	20.55 (3.43)	19.29 (3.85)	16.07 (3.87)	16.36 (3.86)	16.81 (4.86)	15.90 (4.31)	9.80 (5.33)	11.60 (3.34)	2.53	0.115	31.24	<0.001 *	1.59	0.211
Biceps SK, mm	12.57 (4.62)	11.25 (4.40)	14.64 (5.30)	13.29 (3.75)	10.81 (3.09)	10.62 (4.09)	10.00 (3.86)	9.70 (3.33)	6.82	0.01 *	1.1	0.296	1.02	0.315
Subscapular SK, mm	17.00 (6.54)	14.55 (4.66)	15.00 (3.09)	15.79 (3.26)	20.9 (4.65)	17.14 (4.52)	15.30 (3.43)	16.70 (2.75)	17.62	<0.001 *	1.68	0.198	0.02	0.876
Supraspinal SK, mm	19.05 (6.40)	18.32 (5.34)	18.71 (4.89)	18.64 (4.99)	22.81 (4.88)	22.00 (4.51)	21.30 (4.16)	20.50 (4.09)	2.3	0.133	0.25	0.621	2.3	0.133
Suprailiac SK, mm	18.43 (6.07)	16.89 (5.98)	18.86 (5.32)	17.5 (4.72)	21.57 (5.86)	19.52 (6.78)	20.60 (5.10)	18.50 (4.67)	12.15	<0.001 *	0	0.966	0.64	0.427
Arm relaxed circ, cm	28.38 (3.55)	28.96 (3.42)	25.70 (3.13)	26.08 (3.08)	30.08 (1.97)	30.13 (3.48)	28.83 (2.21)	29.23 (2.03)	4.78	0.031 *	8.79	<0.01 *	0.14	0.712
Arm stretched circ, cm	29.44 (3.42)	29.54 (3.40)	26.23 (3.14)	26.60 (3.12)	32.09 (2.42)	31.87 (2.46)	30.26 (2.11)	28.65 (7.01)	0.16	0.689	15	<0.001 *	0	0.973
Waist circ, cm	79.66 (11.58)	78.48 (12.17)	80.49 (8.27)	78.79 (6.98)	96.56 (9.53)	95.14 (9.42)	95.69 (6.07)	94.85 (4.65)	4.09	0.046 *	0	0.953	1.04	0.309
Hip circ, cm	101.94 (8.69)	100.72 (8.78)	98.53 (7.28)	97.45 (6.17)	104.68 (4.52)	104.46 (4.44)	100.69 (5.29)	99.84 (4.15)	9.37	<0.01 *	4.68	0.033 *	0	0.981
Medial calf SK, mm	17.63 (3.44)	16.58 (6.00)	17.88 (7.24)	15.43 (5.17)	12.57 (4.32)	12.68 (7.00)	11.30 (2.87)	12.00 (1.05)	2.2	0.141	0.41	0.522	0.56	0.458
Lateral calf SK, mm	17.54 (3.61)	16.76 (6.10)	13.64 (4.65)	13.57 (3.99)	12.57 (4.32)	12.68 (7.00)	9.50 (2.27)	10.00 (2.11)	0.47	0.493	11.71	<0.001 *	3.01	0.086
Calf circ, cm	36.58 (2.62)	35.60 (5.08)	31.43 (7.10)	34.14 (3.41)	38.15 (3.02)	37.10 (6.40)	36.75 (1.92)	36.92 (1.76)	0.48	0.488	8.18	<0.01 *	0.04	0.84
Resistance, Ω	523.15 (73.63)	510.87 (66.39)	579.56 (78.25)	555.60 (82.39)	488.03 (52.69)	473.74 (74.65)	452.00 (38.43)	442.72 (41.92)	2.48	0.119	1.91	0.171	0.1	0.756
Reactance, Ω	56.67 (6.33)	55.73 (9.00)	60.46 (11.93)	52.96 (6.99)	52.52 (9.98)	57.11 (8.97)	57.37 (8.52)	57.23 (10.23)	0.84	0.361	0.44	0.511	9.25	<0.01 *
Phase angle, °	6.28 (0.94)	7.03 (1.50)	5.82 (0.53)	5.51 (0.59)	6.21 (1.28)	7.49 (1.23)	7.27 (0.88)	7.44 (1.12)	12.42	<0.001 *	2.5	0.118	8.76	<0.01 *
HGS right, kg	25.77 (5.98)	26.05 (5.75)	22.79 (4.61)	23.93 (4.48)	42.60 (10.13)	42.05 (9.72)	40.20 (7.94)	40.20 (7.64)	0.17	0.682	2.47	0.119	3.48	0.065
HGS left, kg	25.30 (4.9)	25.43 (5.65)	23.64 (4.38)	24.00 (4.24)	41.90 (8.54)	41.29 (8.61)	38.00 (6.7)	38.70 (6.58)	0.02	0.893	2.65	0.106	1.54	0.217
Squat, n	15.73 (3.77)	17.41 (3.69)	14.36 (2.76)	18.50 (3.92)	14.33 (3.80)	17.52 (3.63)	22.00 (6.73)	24.10 (5.53)	28.49	<0.001 *	9.29	<0.01 *	0.24	0.625
Walk, meters	541.07 (70.07)	578.82 (66.21)	460.61 (50.77)	481.61 (63.52)	534.74 (93.65)	564.02 (66.45)	592.00 (48.76)	644.00 (82.13)	28.06	<0.001 *	3.28	0.073	0.24	0.628
%F, %	36.33 (3.89)	34.95 (3.99)	36.03 (2.91)	35.76 (2.46)	31.13 (3.43)	29.47 (3.62)	27.25 (3.15)	27.49 (2.86)	18.15	<0.001 *	1.36	0.246	2.46	0.119
FM, kg	24.15 (6.35)	23.15 (6.38)	21.72 (4.95)	22.87 (6.21)	26.42 (4.62)	24.78 (4.23)	21.49 (3.99)	21.05 (3.90)	4.96	0.028 *	3.7	0.057	0.38	0.537
FFM, kg	41.50 (5.90)	42.19 (6.24)	38.16 (6.08)	40.54 (7.96)	58.16 (5.77)	59.08 (5.28)	56.91 (4.03)	55.15 (4.73)	3.33	0.071	3.67	0.059	1.16	0.285
TUA, cm^2^	65.10 (16.67)	67.67 (16.21)	60.84 (25.88)	70.22 (12.95)	72.32 (9.21)	73.19 (16.12)	89.60 (11.02)	89.40 (9.25)	5.64	0.019 *	2.62	0.109	0.62	0.433
UMA, cm^2^	38.99 (11.35)	42.79 (11.04)	21.52 (6.50)	23.41 (8.05)	49.36 (8.81)	51.65 (10.62)	18.19 (3.97)	19.35 (2.66)	16.07	<0.001 *	107.61	<0.001 *	0.66	0.419
UFA, cm^2^	26.11 (6.72)	24.88 (7.60)	18.59 (4.65)	19.22 (4.81)	22.96 (6.47)	21.53 (8.27)	13.30 (7.27)	15.82 (4.65)	0.9	0.344	25.06	<0.001 *	0.87	0.352
TCA, cm^2^	107.06 (15.48)	105.18 (18.13)	82.36 (30.20)	93.63 (18.79)	116.55 (18.43)	112.66 (28.97)	107.79 (10.97)	108.75 (10.10)	0.01	0.918	11.46	0.001 *	0.06	0.809
CMA, cm^2^	74.92 (11.76)	71.72 (21.91)	60.84 (25.88)	70.22 (12.95)	90.07 (14.19)	89.10 (26.13)	89.60 (11.02)	89.40 (9.25)	0.13	0.721	1.78	0.185	0.59	0.444
CFA, cm^2^	32.14 (6.01)	27.72 (9.51)	21.52 (6.50)	23.41 (8.05)	26.49 (8.22)	23.56 (7.63)	18.19 (3.97)	19.35 (2.66)	8.98	<0.01 *	23.56	<0.001 *	1.62	0.207

Notes: SD, standard deviation; F, Snedecor–Fisher test value; *p*, *p*-value; BMI, body mass index; SK, skinfold thickness; circ, circumference; HGS, hand grip strength; %F, percentage of fat mass; Fm, fat mass; FFM, fat-free mass; TUA, total upper area; UMA, upper muscle area; UFA, upper fat area; TCA, total calf area; CMA, calf mass area; CFA, calf fat area; ♀, female; ♂, male; *, statistically significant.

## Data Availability

Data are available on request from the first author.

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
