# Peer review of "Evaluation of the Effectiveness of a Nordic Walking and a Resistance Indoor Training Program: Anthropometric, Body Composition, and Functional Parameters in the Middle-Aged Population"

_jfmk, 2023, doi:10.3390/jfmk8020079_

Round 1

Reviewer 1 Report

Abstract:

·        “longitudinal study” is generally used for observational studies. This study is an intervention.

·        Line 17, Please add “n=” in front of 70.

·        Please describe outcome measures more in details.

·        Please spell out “BIVA”.

·        Conclusion sentence should be modified. It is very ambiguous to say “many potential benefits”. Additionally, while in sentences for results there were no report on physical activity, the conclusion states that “to remain active and to prevent sedentary behaviors”. Readers can not know what results brought this conclusion sentence only reading the abstract.
Also, it is very hard to understand what brought the authors think “NW seems to be more effective”.

Introduction:

·        1st paragraph describes physical activity epidemiology a bit too much. This paragraph needs to be refined only to describe what is the background for this study.

·        The authors should more describe the rational of this study design; i.e., what brought them think it is important to compare outdoor/indoor activities? What is your hypothesis? Why did the authors select resistance exercise rather than aerobic exercise like treadmill walking as the indoor exercise. Because NW is an aerobic exercise, it is more plausible to compare outdoor with indoor exercise in the context of aerobic exercise. While the authors examined sex-specific analysis, they did not describe why they examined sex differences in this study.

·        When I read the manuscript, I do not see why they did not measure physical activity behaviors, but measured body composition and anthropometric measures. These parameters just reflects one aspect of physical fitness, but in the introduction and discussion sections, what they wanted to clarify was to promote physical activity especially in green space.

Methods:

·        This is an intervention study. “Longitudinal study” is generally used for observational studies.

·        Please provide the rational of sample size.

·        Line 115, Did you check the intensity, heart rate, or RPE, etc.?

·        It is ambiguous to say “the same kind of” here. Please describe more detail. Readers can not understand the protocol.

·        How did you instruct daily life out of the exercise program? Since the motivation of this study is to increase physical activity, it is interesting to evaluate physical activity during the intervention period.

·        Line 200, p<0.05.

Results:

·        Line 206, Is this value (61.3+/-8.37) for men or women?

·        Please report adherence of each exercise group.

Discussion:

·        First paragraph is too long, and most part of 1st paragraph should be described in the introduction section if they think it is important to describe.

·        Other paragraphs are also a bit too long as well.

Author Response

Abstract:

· “longitudinal study” is generally used for observational studies. This study is an intervention.

Answer (A): Thank you for the observation, the term was fixed.

· Line 17, Please add “n=” in front of 70.

A: The term was added.

· Please describe outcome measures more in details.

A: The information was added.

· Please spell out “BIVA”.

A: The spell of “BIVA” was added

· Conclusion sentence should be modified. It is very ambiguous to say “many potential benefits”. Additionally, while in sentences for results there were no report on physical activity, the conclusion states that “to remain active and to prevent sedentary behaviors”. Readers can not know what results brought this conclusion sentence only reading the abstract. Also, it is very hard to understand what brought the authors think “NW seems to be more effective”.

A: The conclusions were rewritten.

Introduction:

· 1st paragraph describes physical activity epidemiology a bit too much. This paragraph needs to be refined only to describe what is the background for this study.

A: The paragraph was refined.

· The authors should more describe the rational of this study design; i.e., what brought them think it is important to compare outdoor/indoor activities? What is your hypothesis? Why did the authors select resistance exercise rather than aerobic exercise like treadmill walking as the indoor exercise. Because NW is an aerobic exercise, it is more plausible to compare outdoor with indoor exercise in the context of aerobic exercise. While the authors examined sex-specific analysis, they did not describe why they examined sex differences in this study.

A: The information were added in lines 72-76 and 82-83.

· When I read the manuscript, I do not see why they did not measure physical activity behaviors, but measured body composition and anthropometric measures. These parameters just reflects one aspect of physical fitness, but in the introduction and discussion sections, what they wanted to clarify was to promote physical activity especially in green space.

A: The goal of the article is to evaluate the efficacy of two different kinds of physical activity, indoor resistance training and Nordic walking. To assess this, we measured anthropometric characteristics, body composition parameters and physical tests.

Methods:

· This is an intervention study. “Longitudinal study” is generally used for observational studies.

A: The term was fixed.

· Please provide the rational of sample size.

A: We added it in lines 110-112.

· Line 115, Did you check the intensity, heart rate, or RPE, etc.?

A: Information were added in lines 131-134.

· It is ambiguous to say “the same kind of” here. Please describe more detail. Readers can not understand the protocol.

A: More information were added.

· How did you instruct daily life out of the exercise program? Since the motivation of this study is to increase physical activity, it is interesting to evaluate physical activity during the intervention period.

A: The motivation of the present study was not to increase the physical activity, but to evaluate the difference in anthropometric characteristics, body composition and physical test in two different kinds of activity.

· Line 200, p<0.05.

A: The “p” was added.

Results:

· Line 206, Is this value (61.3+/-8.37) for men or women?

A: The value is for men. The main age of women is report in the previous line.

· Please report adherence of each exercise group.

A: We added figure 1, which explains how many participants complete the PA plan.

Discussion:

· First paragraph is too long, and most part of 1st paragraph should be described in the introduction section if they think it is important to describe.

A: The paragraph was fixed.

· Other paragraphs are also a bit too long as well.

A: The paragraphs were fixed.

Reviewer 2 Report

Thank you for submitting your valuable manuscript to this journal.

This is a quasi-experimental study on 102 middle-aged participants who engaged in the intervention protocols for three months. The aim of the study was to find the best type of physical activity, in particular Nordic walking (NW) and indoor resistance training, to achieve the most efficient improvement in body composition and physical parameters. Anthropometric assessment, body composition and some physical tests were measured before and after the training period.

There are some important points that need to be considered in the study manuscript:

1-      In the material and methods section (line 104), how and where did you find the participants?

2-      How did you allocate the participants to the intervention groups? Why are there 77 people in NW group and only 25 people in indoor training group?

3-      For the inclusion criteria (line 97), you have to describe the characteristics of the study participants who can enroll, then you must determine some characteristics which cause to exclude them, in the exclusion criteria. So, a chronic disabling disease is an exclusion criteria.

4-      How many men were there in each intervention group? According to the results (line 205), a total of 31 men (36.7%) participated in this study. Since the total number of people in indoor group were 25, probably only 10 men and 15 women were in the indoor group. With this significant difference between the number of men and women between the intervention groups, the subgroup analysis based on the sex differences (table 2) is subject to bias. This is the reason for discrepancy between the results of body composition analysis in the whole participants and BIVA analysis in men and women.
In the whole group analysis, both NW and indoor groups had significant results on FM and %F, but in BIVA analysis, significant differences were only in indoor trained women and only NW trained men. So, please remove table 2 from the results of this study and report the BIVA analysis of the whole participants in the intervention groups. You can show the R-Xc and paired graphs for the multivariate changes in classic resistance and reactance of the whole participants in the indoor and NW groups.

5-      According to the results, there is no difference in the physical tests, weight and BMI between groups, and just a few anthropometric measures have negligible differences. Moreover, FM and %F have significant and very large differences at baseline between groups which make the analysis of differences between groups incorrect. Therefore, please revise the conclusion of the study (line 169) and emphasize the positive results of both training protocols on the outcomes of the study. Please revise the discussion section accordingly.

6-      For the title of the manuscript, please add indoor “resistance” training program.

7-      Abstract (line 16), 102 participants including 77 NW and 31 indoor training?!

8-      On introduction (line 31), Please revise the definition of sedentary behavior. Activities with energy expenditure of less than 1.5 METs are low intensity activities. Sedentary behavior is a kind of behavior and not an activity!

9-      Please explain more about the intensity of the intervention programs on section 2.2 of the methods.

10-  Please edit “supraspinal” skinfold to “supraspinale” throughout the manuscript.

Good luck.

The manuscript needs serious editing in terms of scientific writing and English language.

Author Response

Thank you for submitting your valuable manuscript to this journal.

A: Dear reviewer, thank you for your comment and suggestions.

This is a quasi-experimental study on 102 middle-aged participants who engaged in the intervention protocols for three months. The aim of the study was to find the best type of physical activity, in particular Nordic walking (NW) and indoor resistance training, to achieve the most efficient improvement in body composition and physical parameters. Anthropometric assessment, body composition and some physical tests were measured before and after the training period.

There are some important points that need to be considered in the study manuscript:

1- In the material and methods section (line 104), how and where did you find the participants?

Answer (A): Information about recruitment was added in lines 93-96.

2- How did you allocate the participants to the intervention groups? Why are there 77 people in NW group and only 25 people in indoor training group?

A: The two-sport societies allow participants the opportunity to select which kind of PA carry out. However, we added further information in chapter “study design and participants”.

3- For the inclusion criteria (line 97), you have to describe the characteristics of the study participants who can enroll, then you must determine some characteristics which cause to exclude them, in the exclusion criteria. So, a chronic disabling disease is an exclusion criteria.

A: The exclusions criteria were added.

4- How many men were there in each intervention group? According to the results (line 205), a total of 31 men (36.7%) participated in this study. Since the total number of people in indoor group were 25, probably only 10 men and 15 women were in the indoor group. With this significant difference between the number of men and women between the intervention groups, the subgroup analysis based on the sex differences (table 2) is subject to bias. This is the reason for discrepancy between the results of body composition analysis in the whole participants and BIVA analysis in men and women. In the whole group analysis, both NW and indoor groups had significant results on FM and %F, but in BIVA analysis, significant differences were only in indoor trained women and only NW trained men. So, please remove table 2 from the results of this study and report the BIVA analysis of the whole participants in the intervention groups. You can show the R-Xc and paired graphs for the multivariate changes in classic resistance and reactance of the whole participants in the indoor and NW groups.

A: Table 2 was removed, and the graphs were fixed.

5- According to the results, there is no difference in the physical tests, weight and BMI between groups, and just a few anthropometric measures have negligible differences. Moreover, FM and %F have significant and very large differences at baseline between groups which make the analysis of differences between groups incorrect. Therefore, please revise the conclusion of the study (line 169) and emphasize the positive results of both training protocols on the outcomes of the study. Please revise the discussion section accordingly.

A: The discussion part was fixed. (mancano le aggiunte relative alla body composition)

6- For the title of the manuscript, please add indoor “resistance” training program.

A: The title was fixed.

7- Abstract (line 16), 102 participants including 77 NW and 31 indoor training?!

A: The information was fixed.

8- On introduction (line 31), Please revise the definition of sedentary behavior. Activities with energy expenditure of less than 1.5 METs are low intensity activities. Sedentary behavior is a kind of behavior and not an activity!

A: The introduction was fixed according to the suggestion of the other reviewer.

9- Please explain more about the intensity of the intervention programs on section 2.2 of the methods.

A: The Information was added.

10- Please edit “supraspinal” skinfold to “supraspinale” throughout the manuscript.

A: The term was fixed in the manuscript.

Round 2

Reviewer 2 Report

Thank you for revising your valuable manuscript.

I have no further comments on the paper.

Good luck.

The English language is fine and only minor editing is needed.